# Modelling the effects of diurnal temperature variation on malaria infection dynamics in mosquitoes
Isaac J. Stopard [1] ✉, Antoine Sanou [2,3], Eunho Suh [4], Lauren J. Cator [5], Matthew B. Thomas [4,6,7], W. Moussa Guelbéogo [2], N'Falé Sagnon [2], Ben Lambert [8,9] & Thomas S. Churcher [1,9]

Mosquito infection experiments that characterise how sporogony changes with temperature are increasingly being used to parameterise malaria transmission models. In these experiments, mosquitoes are exposed to a range of temperatures, with each group experiencing a single temperature. Diurnal temperature variation can, however, affect the sporogonic cycle of *Plasmodium* parasites. Mosquito dissection data is not available for all temperature profiles, so we investigate whether mathematical models of mosquito infection parameterised with constant temperature thermal performance curves can predict the effects of diurnal temperature variation. We use this model to predict two key parameters governing disease transmission: the human-to-mosquito transmission probability and extrinsic incubation period – and, embed this model into a malaria transmission model to simulate sporozoite prevalence with and without the effects of diurnal and seasonal temperature variation for a single site in Burkina Faso. Simulations incorporating diurnal temperature variation better predict changes in sporogony in laboratory mosquitoes, indicating that constant temperature experiments can be used to predict the effects of fluctuating temperatures. Including the effects of diurnal temperature variation, however, did not substantially improve the predictive ability of the transmission model to predict changes in sporozoite prevalence in wild mosquitoes, indicating further research is needed in more settings.

Thermal performance curves (TPCs) for certain malaria-causing *Anopheles* mosquito and *Plasmodium falciparum* traits indicate they are strongly temperature-dependent[1–3]. During sporogony mosquitoes ingest parasite gametocytes when they feed on an infectious host, the parasites then undergo sexual and asexual reproduction via multiple life-stage transitions, culminating in the infectious sporozoite life-stage in the mosquito salivary glands[4]. These processes determine two important quantities for transmission: (1) the maximum proportion of mosquitoes that are infected following blood-feeding on an infectious host (human-to-mosquito transmission probability [HMTP]) and (2) the delay between mosquito infection and infectiousness (the extrinsic incubation period [EIP]). To transmit malaria, mosquitoes must survive the EIP, which is long relative

to mosquito life expectancy, meaning theoretically the proportion of mosquitoes that are infectious (sporozoite prevalence) is highly sensitive to changes in the EIP and mosquito survival[5]. Laboratory experiments have demonstrated that the EIP declines with temperature[1–3], and the HMTP and mosquito survival have unimodal relationships, with both peaking at intermediate temperatures[1,6–9].

Laboratory-derived TPCs have been used to estimate how temperature affects malaria transmission. Embedding the temperature dependencies of certain entomological parameters within the basic reproduction number ($R_0$) equation, for example, indicates a unimodal relationship between ambient temperature and transmission, which peaks between 25 °C and 33 °C[10,11]. To predict ecological responses to

[1]MRC Centre for Global Infectious Disease Analysis, School of Public Health, Faculty of Medicine, Imperial College London, London, UK. [2]Centre National de Recherche et de Formation sur le Paludisme (CNRFP), Ouagadougou, Burkina Faso. [3]Université Yembila-Abdoulaye-Toguyeni (UYAT), Fada N'Gourma, Burkina Faso. [4]Center for Infectious Disease Dynamics, Department of Entomology, Penn State University, University Park, PA, USA. [5]Department of Life Sciences, Imperial College London, Silwood Park, Ascot, UK. [6]York Environmental Sustainability Institute, Department of Biology, University of York, York, UK. [7]Invasion Science Research Institute and Department of Entomology and Nematology, University of Florida, Gainesville, FL, USA. [8]Department of Statistics, University of Oxford, Oxford, UK. [9]These authors contributed equally: Ben Lambert, Thomas S. Churcher. ✉e-mail: isaac.stopard11@imperial.ac.uk

global heating TPCs are often used to estimate either the instantaneous trait values associated with a future temperature or the average trait values given a range of temperatures[12]. Indeed, malaria transmission models that incorporate laboratory-estimated TPCs are increasingly being used to investigate the role of shifting temperature regimes on malaria transmission[13–16]. Whether malaria incidence data indicates that temperature increases in high altitude, epidemic settings have caused larger malaria outbreaks is, however, debated[17–22]. And, less is known about the importance of temperature in high transmission settings where small changes in the entomological inoculation rate (EIR) have little effect on human malaria prevalence[23]. Malaria trends in endemic settings are strongly influenced by other factors such as human immunity and the use of control interventions[24], which may obfuscate the relationship between temperature and transmission[25,26]. Identifying whether TPCs can be used to predict infection dynamics in mosquitoes, rather than people, may therefore help identify how temperature affects transmission intensity.

Evaluating of the use of laboratory-derived TPCs is required for multiple reasons. First, the laboratory experiments typically measure changes in these parameters when mosquitoes and parasites are exposed to constant temperatures[1–3]. And, to incorporate temperature-dependency in the EIP in malaria transmission models, the degree-day model estimate[3] corresponding to a single daily or monthly temperature is typically used[13,27]. These traits might, however, also vary due to differences in the diurnal temperature variation[28]. Studies exploring the effects of temperature fluctuations on the EIP apply a rate-summation approach[28,29], which predicts that, at low mean temperatures, temperature fluctuations decrease the EIP and vice versa at high mean temperatures[28]. These predictions have been borne out in laboratory studies at low mean temperatures[30] or with the rodent malaria parasite: *Plasmodium chabaudi*[31]. The EIP values predicted using this approach are, however, affected by whether the temperatures used are averaged over hours, days or months[32]. And, evidence from other species indicates performance in fluctuating temperatures might not correspond directly to that in the corresponding constant temperatures due to differences in phenotypic plasticity and acclimation[33], and lagged effects of exposure to thermal extremes[34]. The HMTP, for example, is determined during the early stages of parasite development[9], meaning that applying the HMTP TPC with mean daily or monthly temperatures may not capture the importance of temperature fluctuations. In the laboratory, for example, differences in the HMTP have been observed due to different combinations of diurnal temperature variation and time of mosquito blood feeding[35]. Whether the effects of diurnal temperature variation on the EIP and HMTP of *P. falciparum* in the laboratory can be predicted using TPCs estimated from mosquitoes maintained at constant ambient temperatures therefore needs to be assessed[36]. Finally, microclimatic differences in temperature, such as differences in indoor and outdoor diurnal temperature fluctuations exist[37]. The temperatures mosquitoes are exposed to, and the consequential impact on the EIP and HMTP, might therefore depend on mosquito resting behaviour and differences in microclimate.

In this study, we explore whether it is possible to extrapolate from the results of constant temperature experiments to those where temperatures fluctuate diurnally. To do so, we develop a mathematical model of mosquito infection dynamics, which specifically allows for the fine scale effects of temperature on the EIP and HMTP. We show how this model, trained on published EIP and HMTP TPCs from constant temperature experiments with *Anopheles gambiae* and *P. falciparum*[1], can be used to predict mosquito infection in laboratory experiments (standard membrane feeding assays [SMFAs]) when mosquitoes are exposed to diurnal temperature variation. We then explore how microclimatic temperature data affects the predicted seasonality in the EIP and HMTP using this model, compared to using the TPC values at the mean temperatures. Finally, these temperature-dependent parameter estimates are incorporated into the classic malaria transmission model[38,39], and the model predicted sporozoite prevalence is compared to those sampled in the Cascades region of Burkina Faso; a hot area with high malaria transmission.

## Results

### EIP and HMTP estimates from constant temperature experiments can be used to predict the outcomes of experiments with diurnal temperature variation

Laboratory data from multiple studies show that temporal changes in the infection of *A. gambiae* with *P. falciparum* varies between mosquitoes maintained with different temperature profiles, which can vary due to differences in the diurnal temperature range, mean temperature and biting time (Fig. 1). These studies were conducted in the same laboratory using similar methods, though there were differences in the gametocyte density of the blood-feeds[1,30,40]. SMFA mosquito dissection data are not available for all possible temperature profiles observed in field settings. EIP and HMTP TPCs have, however, been estimated across a wide range of temperatures from experiments where mosquitoes were maintained at constant temperatures (Fig. 2). We therefore developed a mathematical model that explicitly captures within-mosquito infection dynamics in SMFAs: uninfected mosquitoes are blood-fed at time zero, with a temperature-dependent proportion of these becoming infected and transitioning through a series of exposed sub-stages according to a time-dependent parasite development rate (PDR), which varies according to fine-scale changes in temperature. And, investigated whether this model can be parameterised using EIP and HMTP TPCs to predict the outcome of fluctuating temperature experiments.

TPCs have been estimated for the EIP and HMTP of *P. falciparum* (NF54 strain) within either *A. gambiae* or *Anopheles stephensi*[41] (Fig. 2). Observed sporozoite prevalence differed between the studies (Fig. 2A) but once differences in parasite induced mosquito mortality are accounted for there is substantial overlap in the EIP, indicating that temperature-dependencies in the EIP are conserved between the different vector species (Fig. 2B). In contrast, the HMTP TPCs differ considerably: there is a strong modal shape for *A. gambiae* but not *A. stephensi* (Fig. 2C), which could partly be due to a lack of data for *A. stephensi* at lower temperatures. For all subsequent analyses we therefore used the *A. gambiae* specific TPCs.

Models parameterised using these TPCs can predict the timing of increases in sporozoite prevalence (indicative of the EIP distribution) and the maximum proportion of infectious mosquitoes (indicative of the HMTP) of mosquitoes that were housed in fluctuating temperatures in the laboratory (Fig. 1). Varying the PDR estimated from the TPC of the mean EIP according to the instantaneous temperature consistently increases the predictive accuracy of the model compared to using the mean EIP given the mean daily temperature (Table 1), and reproduces the observation that at a mean temperature of 19 °C and biting time of Zeitgeiber time 12 (ZT; 0 is the beginning of day and 12 is the beginning of night) fluctuating temperatures reduced the time until infectious mosquitoes were first observed, but not at a mean temperature of 27 °C (Fig. 1).

In the best fitting model the HMTP is predicted by the value corresponding to the HMTP TPC value at the maximum temperature mosquitoes were exposed to during the first approximately 10 hours post blood-feeding (Table 1). This method reproduces the observation that when there are diurnal temperature fluctuations the maximum sampled sporozoite prevalence is reduced if mosquitoes are exposed to hotter temperatures during early infection: at 27 °C with a DTR of 10 °C, for example, the maximum sampled sporozoite prevalence is reduced as the time of infection became closer to the beginning of the day (ZT12 to ZT18 to ZT0) (Fig. 1), which is not replicated in the simulations using the other methods to estimate the HMTP (Figure S1).

### The best fitting model predicts strong temperature-dependent seasonalities in the EIP and HMTP for a high transmission area of Burkina Faso

Field relevant temperature data was used to investigate how temperature might alter the EIP and HMTP in wild mosquitoes. Microclimatic temperature was estimated by placing temperature loggers indoors and outdoors during the primary transmission season. These temperatures were compared to the ERA5 reanalysis temperatures for the grid square

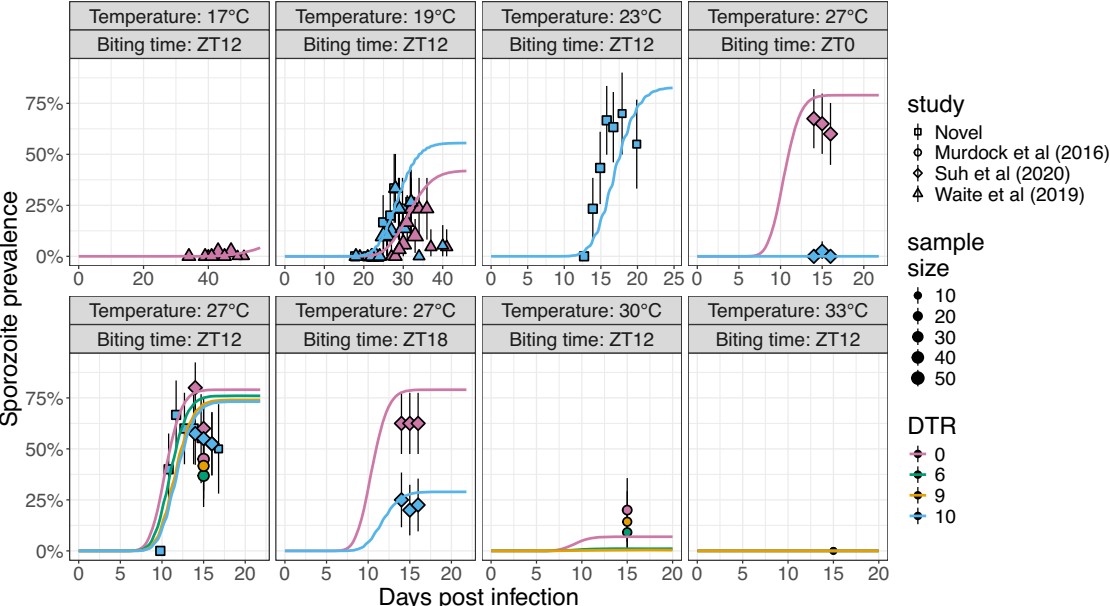

**Fig. 1 | Predicting the effects of different temperature profiles on sporozoite prevalence in experimentally infected mosquitoes.** Sporozoite prevalence (solid coloured lines) was simulated by the best fitting model (model 4; Table 1), in which the extrinsic incubation period (EIP) was determined by temporal changes in temperature and the human to mosquito transmisision probability (HMTP; $c$) was determined by the maximum temperature during the first 10 hours. The points show the sampled sporozoite prevalence values and the vertical lines show the 95% confidence interval estimates. A total of $n$ = 2812 mosquitoes from all studies (including[30,35,40] and novel data) were included. The top title for each plot is the central temperature, the lower title is the biting time, the shape of the points indicate the study the data was obtained from, and the colour shows the diurnal temperature range (DTR). For all plots biting time is in Zeitgeiber time (ZT), where 12:00 is the beginning of night time.

containing Tiefora, Burkina Faso, which are estimated by combining historical observations with data assimilation and modelling to provide hourly estimates with a horizontal resolution of 31 km[42]. The daily mean temperatures and DTRs in Tiefora vary substantially throughout the year (Fig. 3A). The ERA5 temperature estimates systematically differed from the microclimatic temperature logger data: in general, the indoor temperatures recorded were warmer than the ERA5 estimates, especially early in the morning between September and November (Fig. 3A). The DTRs recorded indoors and estimated in the ERA5 re-analysis data were less than those recorded outdoors (Figures S2 & S3).

Using the model that best predicted the laboratory data (model 4; Table 1; henceforth denoted DTR-dependent model) with these temperatures resulted in seasonal differences in the predicted EIP (Fig. 3B) and HMTP (Figure 3C). We estimated changes in the EIP and HMTP given mosquitoes are infected at different times, and calculated the expected daily values using empirical mosquito biting time data from Burkina Faso[43]. The predicted daily EIP increases throughout the wet season (approximately April to October): using the ERA5 2m air temperature data the minimum predicted $EIP_{50}$ is 9.4 days, which occurs in April, and the maximum predicted $EIP_{50}$ was 14.3 days, which occurs at the end of August (Fig. 3B). Similar seasonal trends are observed when predicting the EIP with the microclimatic temperature data; these estimates are, however, shorter (Figure 3B), reflecting the higher temperatures observed on the temperature loggers. The absolute estimates vary depending on the primary mosquito resting location (indoors or outdoors) (Fig. 3B): the expected daily EIP is predicted to be shorter for mosquitoes resting exclusively indoors compared to mosquitoes resting exclusively outdoors. There is also strong seasonality in the predicted HMTP values: the predicted daily HMTP substantially increases during the high transmission season (Fig. 3C). The time of mosquito blood feeding has very little effect on the predicted EIP due to its long duration (Figure S4), but is predicted to have a strong impact on the HMTP: with mosquitoes that feed early in the evening predicted to have the highest HMTP (Fig. 4). In this region of Burkina Faso the model predicts that hotter temperatures decrease both the EIP and HMTP.

Using hourly fluctuations in temperature to predict these parameters gives different trends compared to the TPC values corresponding to the mean monthly temperature (henceforth denoted DTR-independent model) for the HMTP, but less so for the EIP (Fig. 3). The mean temperatures in January and August, for example, are very similar, but January has much larger DTRs, meaning the HMTP predicted using the DTR-dependent model is lower than the HMTP predicted using the DTR-independent model in January but not August (Fig. 3B). In contrast, in this hot setting there are little differences between the EIP values predicted by the DTR-dependent and DTR-independent models.

## Accounting for diurnal temperature variation does not substantially improve predictions of malaria infection dynamics in wild mosquitoes

Malariology has a long history of applying mechanistic transmission models to help understand disease dynamics and control[39,44]. The Ross-Macdonald model is widely used in mosquito-borne disease epidemiology: important metrics, such as the basic reproduction number ($R_0$), for example, were derived from this model[5,39] and the model assumptions are common among mosquito-borne disease transmission models[45]. To assess the importance of temperature-dependent changes in the EIP, HMTP and the per-mosquito mortality rate on entomological malaria risk in malaria transmission models, we incorporated the DTR-dependent, DTR independent methods and constant EIP and HMTP parameter estimates into an adapted Ross-Macdonald malaria transmission model[5,38,46]. Annual seasonality in the vector-host ratio was estimated from sampled human biting rate (HBR) data using a generalised-additive model (GAM) (Fig. 5A). Mosquitoes were present year round but the number caught peaked in early October (Fig. 5A). The per-mosquito adult emergence rate in the malaria transmission model was varied seasonally to replicate the GAM modelled seasonality in the indoor HBR (assuming a constant per-mosquito biting rate), given the modelled per-mosquito mortality rate. Temperature-dependent mortality rates were estimated using the Marten's model 2[6]. The first sporozoite positive mosquitoes are detected in May with prevalence peaking

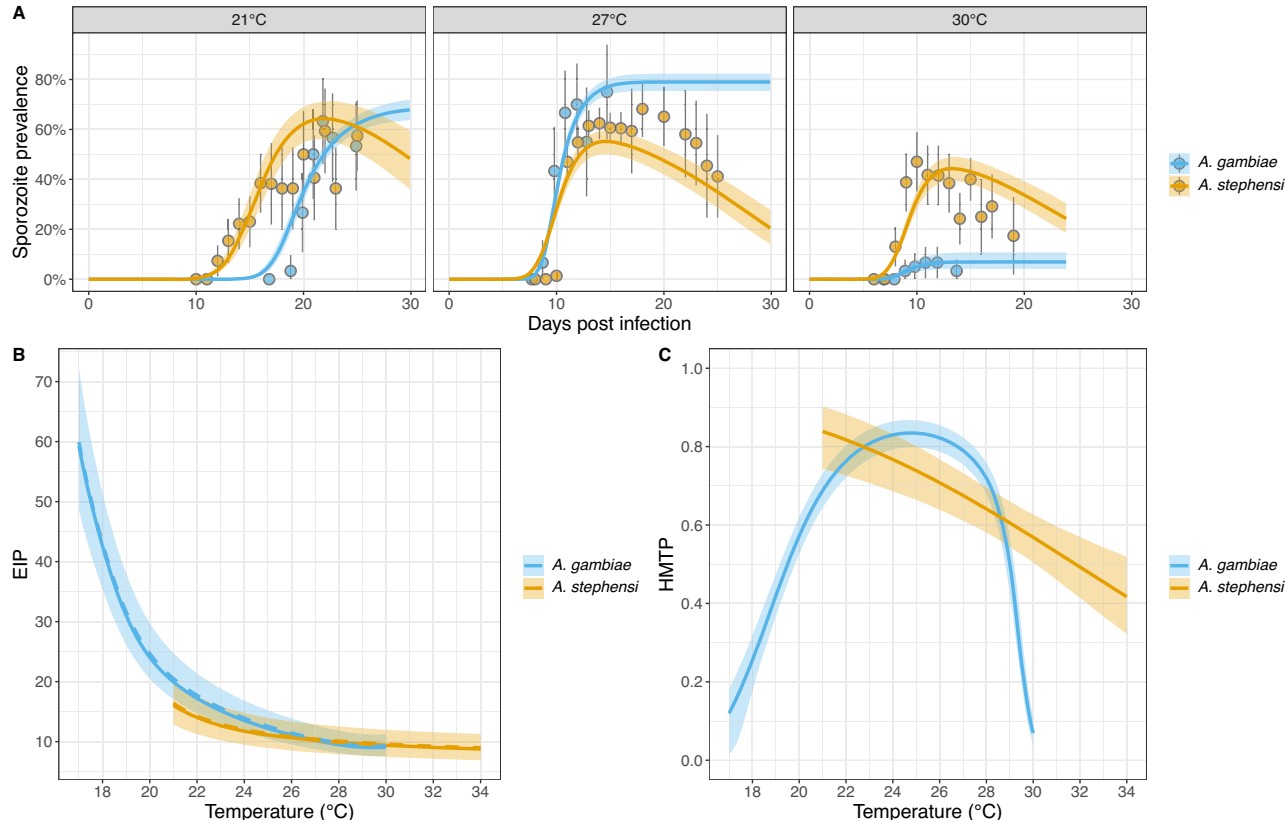

**Fig. 2 | Thermal performance curves for the EIP and HMTP of _P. falciparum_ infected _A. gambiae_ and _A. stephensi mosquitoes._ Thermal performance curves for the EIP and HMTP of _P. falciparum_ infected _A. gambiae_ and _A. stephensi_ mosquitoes.** **A** Observed sporozoite prevalence (points) with 95% confidence intervals and model fits with 95% credible intervals for mosquitoes maintained at different constant temperatures, at temperatures where both species were tested.

**B** EIP thermal performance curves. The solid line is the $EIP_{50}$ estimate, the dashed line is the mean EIP and the shaded area shows the difference between $EIP_{10}$ and $EIP_{90}$. **C** Species specific HMTP thermal performance curves. Uncertainty in the HMTP is shown by the 95% credible intervals. Data were obtained from[2] and[1], and model fits were obtained from Stopard, Churcher & Lambert[41] for _A. stephensi_ and Suh et al.[1] for _A. gambiae_ respectively.

### Table 1 | Predictive accuracy of the different models to simulate the effects of diurnal temperature variations on sporozoite prevalence during SMFAs

| Model name | Method to simulate changes in $\beta$ | Methods to estimate the HMTP | Fitted $h$ | Log-likelihood | AUC |
|---|---|---|---|---|---|
| 1 | Mean | i | * | −737.33 | 0.75 |
| 2 | Fluctuating | i | * | −543.59 | 0.77 |
| 3 | Mean | ii (maximum) | 10.0 | −409.8 | **0.82** |
| **4** | **Fluctuating** | **ii (maximum)** | **10.0** | **−305.93** | **0.82** |
| 5 | Mean | ii (mean) | 0 | −652.47 | 0.77 |
| 6 | Fluctuating | ii (mean) | 0 | −528.83 | 0.78 |
| 7 | Mean | ii (minimum) | 0 | −652.47 | 0.77 |
| 8 | Fluctuating | ii (minimum) | 0.7 | −512.6 | 0.79 |
| 9 | Mean | iii (maximum) | 0 | −652.47 | 0.77 |
| 10 | Fluctuating | iii (maximum) | 0 | −528.83 | 0.78 |
| 11 | Mean | iii (mean) | 23.8 | −543.22 | 0.78 |
| 12 | Fluctuating | iii (mean) | 20.6 | −381.17 | 0.79 |
| 13 | Mean | iii (minimum) | 0.1 | −654.91 | 0.77 |
| 14 | Fluctuating | iii (minimum) | 3.5 | −556.04 | 0.79 |

A range of methods are used to calculate the HMTP ($c$) and the transition rate between exposed mosquito states ($\beta$), which are compared by their ability to predict laboratory sporozoite prevalence given mosquitoes are exposed to different temperature profiles. $h$ is the number of hours post infection that are used to estimate the HMTP. * indicates $h$ was not estimated because the HMTP ($c$) does not vary with $h$ for this model.

in late July (Fig. 5B). We fitted the adapted malaria transmission model with the DTR-dependent, DTR-independent and constant parameter estimates to the temporal changes in the sampled sporozoite prevalence by scaling the HMTP (Fig. 5B, Table S1). The laboratory-derived HMTP estimates were scaled to fit the observed sporozoite prevalence in the field using maximum likelihood (Table S1). Models where the EIP, HMTP and per-capita mosquito mortality rate changed with hourly temperature (DTR-dependent) fit the observed data in some simulations better than models which used mean monthly temperatures (DTR-independent) or took constant values (constant model), but others not (Table S1). For example, assuming a per-mosquito biting rate of $\frac{1}{3}$ and human recovery rate of $\frac{1}{100}$ then the mean absolute error on sporozoite prevalence (MAE) was 2.91% for the DTR-dependent, 3.14% for the DTR-independent and 3.21% for the constant model. The greatest difference between the models was observed at the beginning of the year, when the constant model predicted higher transmission than temperature sensitive models (Table S1). Nevertheless, the similarity of the predictions and noise in the entomological data means it remains unclear whether finer-scale temperature changes substantially improves model fit. The ability of the model to capture changes in sporozoite prevalence over time also depends on other parameters such as the mosquito biting rate, which are highly uncertain. A sensitivity analysis on these parameters shows the differences in predictions between the different models is consistently minimal (Table S1).

## Discussion

This work indicates that the EIP and HMTP TPCs from constant temperature experiments can be used to predict the effects of temperature in SMFAs with fluctuating temperatures. This is important as it could enable a

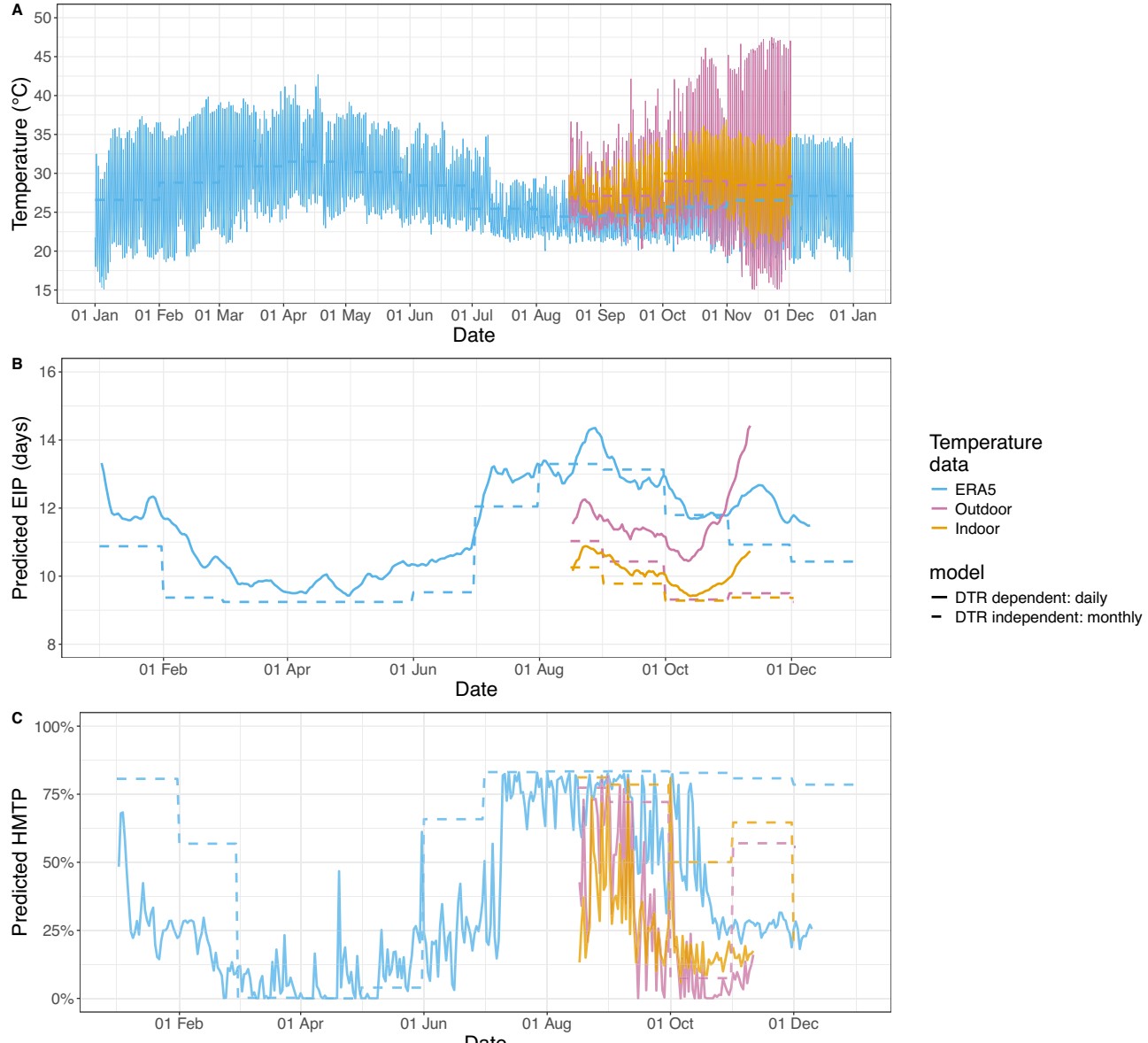

**Fig. 3 | Seasonal and microclimatic temperature-dependent differences in the model-derived EIP and HMTP predicted for the Cascades region of Burkina Faso. A** Hourly changes in temperature for Tiefora, Burkina Faso. Outdoor refers to the hourly temperature logger values outside a single house. Indoor refers to the hourly temperature averaged from temperature loggers placed in six houses. The dashed lines show the mean monthly temperatures. **B** The expected EIP for each day. For the DTR-dependent model the solid lines show the $EIP_{50}$. **C** The expected HMTP for each day. For (**B**, **C**) the dashed line shows the predicted values corresponding the the thermal performance curves at the mean monthly temperatures (DTR-independent model), and the solid line shows the predicted values using the DTR-dependent model (Table 1: model 4), which accounts for hourly temperature changes and differences in the time of mosquito infection. For the DTR-dependent model, daily estimates were calculated as the expected values given differences in the proportions of mosquitoes biting (and being infected) at different times calculated using data from[43]. Temperature, EIP and HMTP values are from the year 2020.

limited set of laboratory experiments to predict transmission in settings with diverse temperature profiles. At lower temperatures, we demonstrated for *P. falciparum* the impact of diurnal temperature variation can be predicted using a temporally fluctuating PDR that varies according to the instantaneous temperature and the constant temperature-EIP relationship. These methods assume the effect of temperature on the PDR occurs over the duration of sporogony. For temperatures lower or greater than the experimental constant temperatures, we assumed the PDR was the lowest or highest values respectively. Experimental data for the impact of DTR at higher temperatures are lacking but would be required to further validate this model (the maximum experimental temperature applied was 32 °C within the 27 °C DTR experiment). Nonetheless, this method improves the model fits of laboratory data and replicates the observation that the EIP of

mosquitoes maintained at 19°C with a DTR of 10 °C is shorter than those with a DTR of 0 °C, while there is little difference at 27 °C. The magnitudes of difference are also closely mirrored. Existing EIP estimates suggest that the PDR does not decline to zero at temperatures below 35 °C[2]. We assumed for temperatures greater than this the PDR did not decline to zero, which differs from assumptions in other studies[10]. It is, however, unclear whether the PDRs do temporally decline to zero, and future research on mosquitoes exposed to higher temperatures during sporogony are required. A reduction in infection and mosquito survival at these temperatures could limit these studies: given our model, it might be possible to estimate the PDR-temperature relationship from mosquitoes that are initially infected at moderate temperatures and exposed to temperature profiles that include extremes. Using the HMTP that corresponds to the maximum temperature

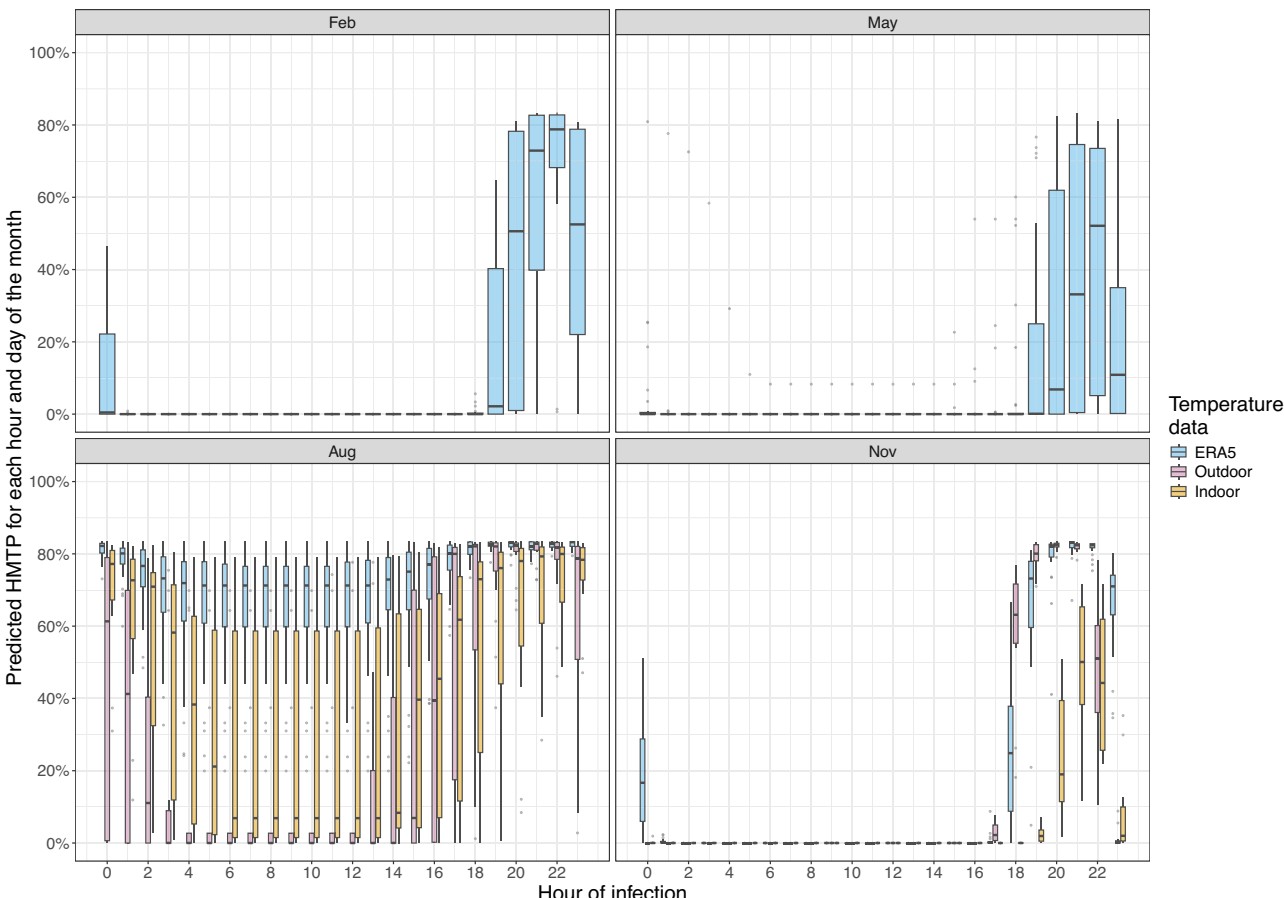

**Fig. 4 | The effects of infection time on the model-derived HMTP.** Outdoor refers to the hourly temperature logger values outside a single house. Indoor refers to the hourly temperature averaged from temperature loggers placed in six houses. Times shown are not in Zeitgeber time. The Tukey-style box plots show the median (horizontal line), 25th percentile (Q1; lower hinge), 75th percentile (Q3; upper hinge), Q1 minus 1.5 times the interquartile range (lower whisker) and Q3 plus 1.5 times this interquartile range (upper whisker) for all predicted HMTP values for different months of the year 2020. Note that for the indoor and outdoor predicted values are only available for the subset of the year when temperature logger data were available.

measured over the first 10 hours of infection correlated best with the observed effect of DTR and biting time on the HMTP. Though, it is unlikely this estimate is completely accurate because SMFAs are noisy and there were differences in the blood gametocyte densities of the different SMFAs[1,30,35,40], our model does, however, allow prediction of the effects of temperature across a range of experiments. And, these findings are consistent with existing evidence that parasite establishment is most sensitive to temperature immediately post infection[35], and early life stages are more sensitive to increases in temperature[9].

The model predicts that the EIP and HMTP in Tiefora vary throughout the year. It also indicates that accounting for diurnal temperature variation can change the HMTP estimates at this site between October and March, with estimates using the mean monthly temperatures substantially overestimating the HMTP. The temperature logger data is hotter than that predicted by ERA5, meaning the HMTP predicted using these data dramatically declines in early October, which is inconsistent with the local entomological data which suggests a relatively high proportion of mosquitoes are infectious at this time of year. Temperature logger data was, however, unavailable for the years of entomological collection, so this will need to be confirmed. These microclimatic differences in temperature could have a substantial impact on mosquito infection in the laboratory, though impact in the field where mosquitoes can move will depend on anopheline mosquito resting and thermal avoidance behaviour[47]. Existing evidence suggests *Anopheles gambiae sensu stricto* are largely endophilic[48] meaning it is necessary to consider the indoor microclimate[47]. Though, a subgroup of *A. gambiae s.s.* and *Anopheles arabiensis* may be more exophilic[48–51]. Housing

design, such as roof type and sources of ventilation can impact indoor temperature[52,53]. Laboratory studies indicate that anopheline thermal avoidance behaviour is limited until temperatures between 33 °C and 35 °C[35,54], and mosquitoes housed along a thermal gradient were found to be non-randomly distributed with the mean resting temperature approximately 26 °C[55]. In the field, *A. gambiae s.l.* have been shown to avoid indoor areas with extreme high temperatures leading to different resting places indoors[56–58] or increased probability of exiting[59]. Irrespective of whether these behaviours are caused by temperature, the framework we have developed could be used to predict mosquito infection in different scenarios. Furthermore, care should be taken interpreting these results. Estimates of the temperature-dependence of important vector and parasite traits for *P. falciparum* transmission were conducted with inbred laboratory parasite and mosquito colony strains[1]. Local parasite strains might be adapted to higher temperatures through either phenotypic plasticity or evolution[60]. *P. falciparum* infection of mosquitoes can vary between sympatric and allopatric parasite-vector combinations[61] and membrane feeding assays indicate that at 32 °C infection of mosquitoes with wild parasites is reduced but still occurs, whereas with NF54 there is no infection[62]. Our work demonstrates these experiments are still, however, valuable for understanding the functional forms of TPCs: the thermal dependency of the EIP, for example, appears more reliable than the HMTP as the patterns are conserved between different mosquito species. The external validity of these experiments might be improved by using local sympatric parasite and mosquito strains, but these models need to be validated before they are used to project future climate impacts.

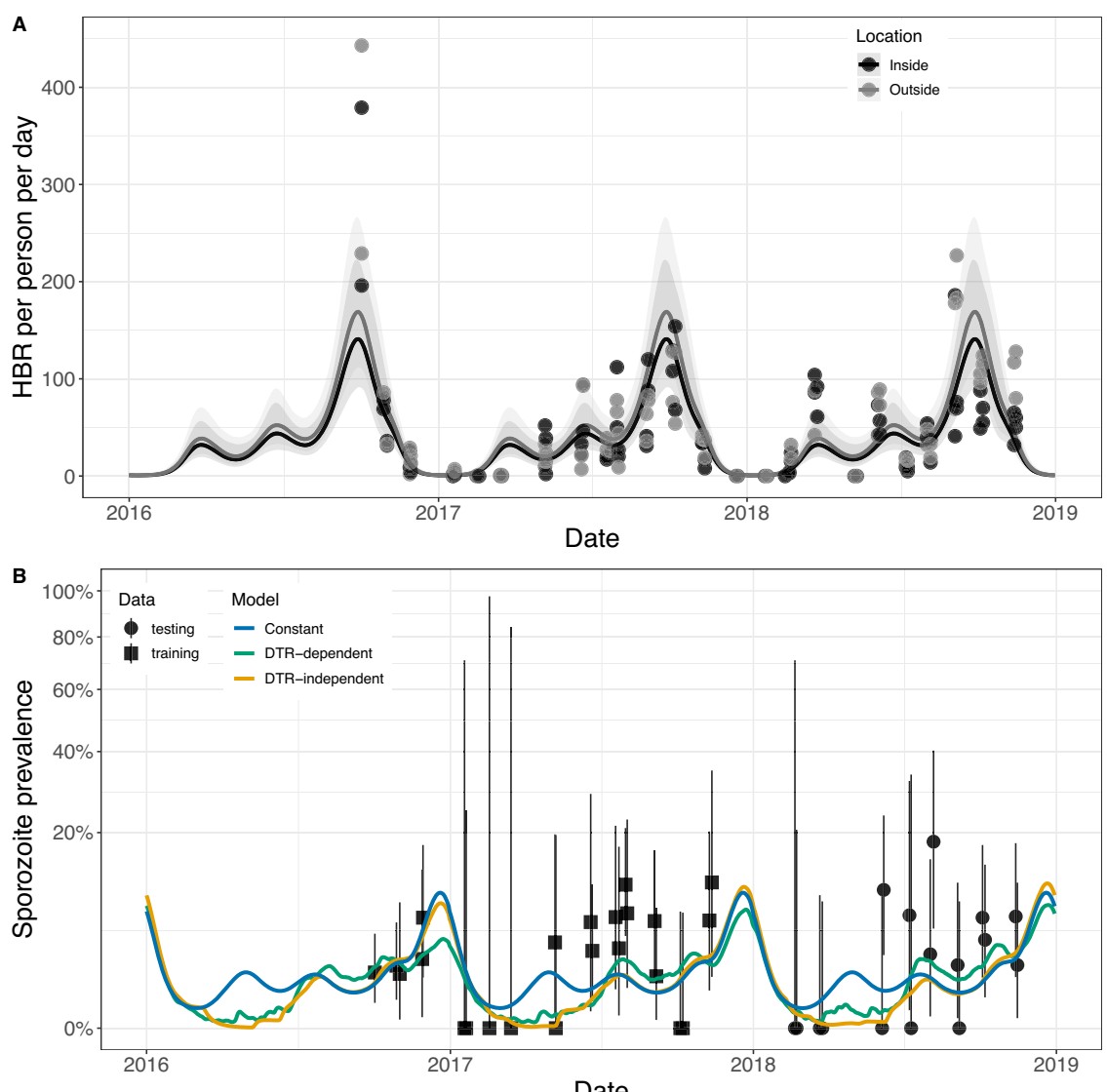

**Fig. 5 | Seasonal differences in the HBR (A) and (B) sporozoite prevalence for the Cascades region of Burkina Faso.** In (**A**) the generalised additive model (GAM) fits are with a univariate cyclic cubic spline with 10 knots for the day of the year. Uncertainty is shown by the 95% credible intervals for the GAM model fit. The adapted Ross-Macdonald models were run with a human recovery rate per person per day of $\frac{1}{100}$, mosquito biting rate of $\frac{1}{3}$ per mosquito per day, and the indoor HBR estimates. For (**A**) the points show the individual sample values, with a total of 8438 mosquitoes included. For (**B**) the uncertainty in the sampled sporozoite prevalence estimates is shown by the Clopper-Pearson 95% binomial confidence intervals, with a total of 1593 mosquitoes tested on all days. Data was obtained from[69].

We attempted to validate the laboratory data and our framework for estimating the EIP and HMTP in a fluctuating environment by including the parameter estimates in a simple transmission dynamics model, and predicting sporozoite prevalence for a single site in Burkina Faso with a specific set of temperatures. The temperature-dependent malaria transmission models were broadly able to capture seasonality. Similarly, the predictions of temperature-dependent $R_0$ models have been shown to correlate with malaria incidence in China[22] and the EIR from a range of settings[10]. Existing transmission dynamics models of malaria[63] assume these parameters are constant throughout the year, which our findings suggest should be reconsidered. The difference between the DTR-dependent and DTR-independent models was modest though. Further data from a range of sites with different temperature profiles is needed to validate the model. The predictive ability of the model would also likely improve with improvements in the transmission dynamics model. Here we used the simple Ross-Macdonald model, but model fits could be improved through using more complex models which capture heterogeneity in transmission and asymptomatic infection, amongst others. For

example, transmission will be influenced by other processes not explored here, such as seasonality in natural human infectiousness and the use of control interventions (specifically insecticide treated nets and seasonal malaria chemoprevention). Indeed, we have not accounted for differences in gametocyte density, though the gametocyte investment in infected people has been shown to increase during the wet season[64]. Increased gametocyte density increases the probability of mosquito infection[65], which can interact with temperature[66], and could therefore partly explain the observed reductions in sporozoite prevalence. Another limiting factor we have not considered is that circadian rhythms in the human blood stage gametocytaemia may mean there are diurnal changes in the infectivity of mammal hosts[67]. Another study did not, however, find a difference in the oocyst number of mosquitoes fed during the day or night[68]. Furthermore, we have assumed any additional mortality due to vector control interventions does not vary seasonally or with time, which is likely to be an oversimplification as vector species composition changes throughout the year[69]. To estimate the effects of temperature on mosquito mortality we used the Martens model, which was estimated

using very few data points[6]. The models assume that the effects of temperature are manifested via the adults alone. However, temperature, as well as seasonal variation in habitat quality and nutrition, will also affect larval development rate and this in turn could spillover to affect adult traits[70]. Small mosquitoes that that might occur at certain times of the year could have different survival[71], biting rates, blood meal size and competence. Identifying field-relevant TPCs for relevant vector species is therefore necessary. The exact causes of the differences in seasonality predicted by our model and those observed in Burkina Faso cannot therefore be determined. The ERA5 temperatures were rarely greater than 35 °C, so we do not believe our assumptions regarding the limits of PDR and HMTP substantially affected our parameter estimates for Tiefora. Whether the ERA5 data is a good predictor of temperatures experienced by the mosquito is, however, unclear, as our empirical temperature estimates showed considerably more variability than ERA5. Sporozoite prevalence data is highly variable and large numbers of mosquitoes are required to differentiate between models unless predictions vary substantially. As previously outlined, the predictions of the DTR-dependent and DTR-independent model were relatively similar in this setting, so the power of the study to differentiate between models is likely to be relatively low. Given the substantial limitations outlined above, we would argue that further research is needed to verify whether daily fluctuations in temperature influence malaria transmission in wild mosquito populations. Understanding discrepancies between the laboratory and different field settings therefore remains critical. Once validated, the framework outlined here could be used to predict mosquito infection in different scenarios, explore how temperature may influence the effectiveness of malaria control interventions in changing environments and incorporated into decision tools for malaria intervention deployment.

## Methods
### Data
**Sporogony data**. SMFAs were conducted to test a mathematical model that aims to explain how parasite infection dynamics within mosquitoes are perturbed in the presence of fluctuating temperatures. Approximately 5-6-day-old laboratory-reared *A. gambiae* (G3 strain) were infected through a standard membrane feeder with *P. falciparum* (NF54 strain; 0.126% gametocytaemia). Mosquitoes were housed at 19 °C, 23 °C, 27 °C with diurnal temperature variations of ±5 °C (DTR: 10 °C). The temperature fluctuated according to the Parton-Logan model, whereby temperatures during the day fluctuated sinusoidally and declined exponentially through the night[72], with the parameterisation following[40] (Fig. S5). A 12h light: 12h dark cycle was used with mosquitoes infected at the start of night (ZT12). Mosquitoes were dissected for sporozoites on a range of days post infection (Table S2).

These data were supplemented with SMFA sporozoite prevalence data from previously published studies from the same laboratory using *A. gambiae* mosquitoes and *P. falciparum* (NF54) parasites (Table S2). These studies[30,35,40] present SMFAs where mosquitoes were exposed to a range of temperature profiles due to differences in the mean daily temperature, DTR and the times the mosquitoes were infected relative to the diurnal temperature variation. Mosquitoes from different studies (including those used to estimate the EIP and HMTP TPCs) received blood-meals with different mean gametocyte densities. Study-level differences in infection were, however, not identifiable due to lack of replication with the same temperature profiles.

**Tiefora temperature data**. Indoor and outdoor temperature logger data were collected from a single site in the Cascades region of Burkina Faso (Tiefora, 10°38'N 4°33'W). Temperatures were recorded at 10-minute intervals and averaged over each hour. Indoor temperatures were recorded from six houses and the mean hourly values calculated (all temperatures are shown in Fig. S3). For each temperature logger, hourly data were missing for less than 10% of observations, and we imputed the missing values using the -"interpolate_gaps_hourly" function from the

chillR package[73]. All microclimatic temperature data were from the year 2020. For comparison, 2-metre air temperature estimates for the grid cell containing Tiefora were obtained for 2015, 2016, 2017, 2018 and 2020 from the ERA5 reanalysis dataset[42]. To obtain continuous changes in temperature the hourly temperatures were linearly interpolated using the R "approxfun" function.

**Tiefora entomological data**. Previously published mosquito human landing catch data and field sporozoite prevalence sampling data for Tiefora were obtained from Sanou et al.[69]. Collections were conducted between late 2016 and 2018. A mass distribution of insecticide treated nets was conducted in Burkina Faso in July 2016.

**Using thermal performance curves to predict the empirically observed EIP and HTMP from experiments where there is diurnal temperature variation**
During SMFAs mosquitoes of a similar age are fed infectious blood through a membrane feeder at a single time point, maintained, and then dissected for sporozoites on subsequent days post infection[74]. The time taken for the cumulative sporozoite prevalence to plateau is indicative of the EIP and the level at which it plateaus is indicative of the HMTP[41]. In previous work we developed a stochastic model of sporogony to estimate the EIP and HMTP TPCs from SMFA sporozoite prevalence data, in which mosquitoes are housed at constant temperatures[41]. This model does not allow temporal variation in the temperature within the same experiment. To assess whether the temporal increases in the cumulative sporozoite prevalence from SMFAs where mosquitoes are exposed to diurnal temperature variation can be predicted using the EIP and HMTP TPCs, we developed a ordinary differential equation (ODE) model, which specifically allows time-dependent rates. We adapted the mosquito infection dynamics model from the classic malaria transmission model[5] to track the infection status at time $t$ of a single cohort of infected blood-fed mosquitoes (there is no birth of susceptible mosquitoes), where all mosquitoes were infected at time zero and there is no continual malaria transmission, and allow the transition rates between infection and infectious to vary with time according to the temperature. In the classic malaria transmission model the EIP is described by delay differential equation models[5], meaning the EIP is fixed for all mosquitoes and cannot vary during the delay[75]. We used the Linear Chain Trick (LCT) to allow the deterministic representation of a stochastic model where the EIP is Erlang distributed: the LCT divides the exposed mosquito compartment into $\alpha$ serially connected sub-compartments. In ODE models if there are two compartments and the first flows into the second at a constant rate then the ODE model gives the mean field approximation of an individual-based stochastic model where the transition times of individuals are exponentially distributed. Existing studies indicate the EIP distribution is better described using a gamma distribution[1,41]. To mimic a gamma distribued EIP we selected $\alpha > 1$ sub-compartments with a transition rate between one exposed compartment and the next of $\beta$, which if constant, yields the mean field approximation of the equivalent individual-level stochastic model where the $EIP \sim Erlang(\alpha, \beta)$[76]. This method has been used to simulate infectious disease transmission, including malaria transmission, with gamma distributed latent periods[77,78]. The numbers of susceptible ($S$), exposed ($Y$) and infectious ($Z$) mosquitoes were simulated using a coupled system of ordinary differential equations (ODEs),

$$
\begin{aligned}
\frac{dS}{dt} &= -gS, \\
\frac{dY_1}{dt} &= -\beta(C[t])Y_1 - gY_1, \\
&\cdots \\
\frac{dY_\alpha}{dt} &= \beta(C[t])Y_{(\alpha-1)} - \beta(C[t])Y_\alpha - gY_\alpha, \\
\frac{dZ}{dt} &= \beta(C[t])Y_\alpha - gZ.
\end{aligned}
\tag{1}
$$

At time zero, the initial numbers of $Y_1$ mosquitoes was determined by the HMTP, which we denote $c$; $S(0) = N - cN$, $Y_1(0) = cN$ and $Y_\alpha(0) = 0$, $Z(0) = 0$, where $N = S(0) + Y_1(0) + \ldots + Y_\alpha(0) + Z(0)$ is the mosquito cohort size at time 0. The per-mosquito mortality rate was assumed to be $g = 0.1$ per day, which does not differ between mosquito states, so does not affect the predicted sporozoite prevalence. $C$ is the temperature at time $t$. To parameterise how $\beta$ varies as a function of temperature, we assumed a fixed value of $\alpha$ and selected $\beta$ to give the same mean at each temperature, $C$, as the corresponding mean EIP derived from previous constant temperature studies (denoted $v(C)$)[1]: $\beta(C) = \frac{\alpha}{v(C)}$. $v(C)$ is the posterior median of the previously estimated mean EIP[1]. Similarly, the HMTP is temperature dependent, such that for each temperature $c = \delta(C)$, where $\delta$ is the median posterior HMTP TPC value (estimated in[1]). We set the shape parameter, $\alpha$, to be the integer that best fitted the temperature-dependent changes to variance reported in the constant temperature experiments[1]. To do so, we used non-linear least squares (R stats package: Gauss-Newton algorithm with a start $\alpha$ value of 20): the optimal $\alpha$ was 47 (46.88 with a convergence tolerance of $5.5 \times 10^{-7}$) Figure (S6A). The model predictions of the sporozoite prevalence values in the constant temperature experiments were assessed for the ODE model and compared to the original model used to estimate the TPCs[1]. The discriminatory ability of each model was assessed using the area under the receiver operating receiver operating characteristic (ROC) curves (AUCs), which were calculated using the pROC package[79] in R. Only small differences in the ROC curves were observed (Fig. S6B). Approximating the EIP distribution using an Erlang distribution minimally decreased the AUC from 0.874 (estimated for mSOS) to 0.871. The scaled Brier scores[80,81] minimally decreased from 0.336 (estimated using mSOS) to 0.325, indicating that both models resulted in similar sporozoite prevalence predictions (Fig. S6C). These findings indicate that the EIP distribution is well approximated using an Erlang distribution.

This model can also be simulated with an instantaneous time-varying rate ($\beta$) and consequently time-varying transition times. Within this model, the rate parameter is varied temporally according to the instantaneous temperature as a function of time: $C(t)$. We tested whether this model could predict the sporozoite prevalence in SMFAs, where the instantaneous temperatures mosquitoes were exposed to varied diurnally with precise form determined by a range of experimental characteristics including the central temperature, diurnal temperature range (DTR) and hour of the day when infection occurred (fluctuating temperatue SMFAs) (Fig. 1; Table S2). To do so, we used the same model, which uses the TPCs estimated from experiments where temperatures are constant throughout the SMFA. There was flexibility in our choice of how to incorporate temperature into our model, and we explored multiple scenarios:

- Diurnal temperature variation-independent model. We set $\beta$ to be constant at the value determined by the mean temperature over 24 h, such that $\bar{C} = \int_0^{24} C(t)dt/24$ and $\beta(C[t]) = \alpha/v(\bar{C})$.
- Fluctuating temperature model. $\beta$ varied temporally according to the fluctuating temperature values. In this model $\beta$ varied with time, meaning the EIP distribution could vary with fine scale temporal changes in temperature: $\beta(C[t]) = \frac{\alpha}{v(C[t])}$.

During the SMFAs the diurnal temperature variation was consistent over 24-h periods and set according to the Parton-Logan model. To simulate fine-scale changes in temperature we calculated the temperature every 0.1 h using the Parton-Logan model, and linearly interpolated the temperature between these times (Fig. S5). In the constant-temperature experiments the minimum temperature a cohort of mosquitoes was exposed to was 17 °C, and the maximum, which another cohort were subject to throughout the assay, was 30 °C. It is unclear whether the parasite development rate (PDR; inverse of the EIP) declines to zero at a critical temperature threshold and existing studies have fitted modal functions to this relationship[10,28]. In the laboratory, high temperatures decrease the HMTP[35] and mosquito survival, meaning it is hard to identify the critical temperature threshold for parasite development. For *A. stephensi*, experimental evidence suggests the relationship between temperature and EIP does not increase until at least

34 °C[2,41]. To avoid making extreme extrapolations about the poorly understood parameter limits, we assumed that for $C(t) < 17$ $v(C)$ was the maximum value of its instantaneous value for $C \in [17, 30]$; for $C(t) > 30$ we assumed it was the minimum value across this temperature range.

To investigate if the the HMTP TPC, $\delta(C)$, and temperature profiles mosquitoes are exposed to could help predict the observed sporozoite prevalence in the fluctuating temperature SMFAs, we assumed the HMTP is temperature-sensitive only during the first 24-hours because experimental evidence indicates the ookinete and early life stages of sporogony are the most temperature sensitive[9,35,62]. Within this interval, the exact period during which the HMTP is temperature sensitive is unclear so in some of our model scenarios, we allowed the data to determine how many hours post blood-feeding, $h$, were informative for predicting changes in the HMTP. Because of the uncertainties, we explored a range of ways of allowing changes in temperature to affect the $c$ parameter:

- $c$ was estimated using the mean daily temperature (denoted $\bar{C}$): $c = \delta(\bar{C})$.
- $c$ was estimated by calculating the arithmetic mean, minimum or maximum temperature between infection and $h$ hours post infection and feeding each of these possibilities into the $\delta$ function.
- $\delta(C)$ was calculated for each temperature the mosquitoes were exposed to between infection and $h$ hours post infection and the arithmetic mean, minimum or maximum of these values was used as the value of $c$.

At each time $t$ post infection, sporozoite prevalence was calculated as $z(t) = \frac{Z(t)}{S(t)+Y_1(t)+\ldots+Y_\alpha(t)+Z(t)}$. To estimate the $EIP_{50}$ from this model we estimated the time for the simulated sporozoite prevalence to reach different percentiles of the maximum sporozoite prevalence, which is determined by $c$.

For each combination of methods used to set $\beta$ and $c$, $h$ was fitted independently for each model by maximum likelihood. Some of our choices for setting $c$ depended on setting a continuous parameter, $h$, and we chose this parameter by maximising the likelihood of of fitting the sporozoite prevalence data ($D$, Table S2);

$$L(z|D) = \prod_{k=1}^{K} \prod_{t=1}^{T} \text{binomial}\left(U_{k,t}|z_{k,t}, V_{k,t}\right), \qquad (2)$$

where $t$ is each time of dissection, $k \in [1, K]$ is each experimental temperature profile (due to the combination of biting time, central temperature and DTR), $z$ is the simulated sporozoite prevalence, $U$ is the number of dissected mosquitoes that are infectious and $V$ is the total number of dissected mosquitoes. All ODE models were numerically integrated using the deSolve R package with the default LSODA integrator[82]. To determine the maximum likelihood estimate of $h$, we used the base R "optim" function with the Brent algorithm[83]. To assess the model accuracy, for each fitted model, the AUCs were calculated using the pROC R package[79] (Table 1).

### Predicting the effects of diurnal temperature range on the EIP and HMTP in Burkina Faso

We used the temperatures from the the ERA5 re-analysis dataset and indoor and outdoor microclimate temperature loggers to predict the temperature-dependent seasonality in the EIP and HMTP in Tiefora, Burkina Faso. There were very little differences in indoor temperatures between different huts (Figure S3) so we used the mean hourly temperature across all huts.

**DTR-independent estimates.** For each time, $t$, we calculated the mean monthly temperature: $\bar{C}_m(t)$. We calculated the EIP and HMTPs at each time using the TPCs, such that,

$$EIP_{DTR-independent}(t) = v(\bar{C}_m[t]),$$
$$c_{DTR-independent}(t) = \delta(\bar{C}_m[t]). \qquad (3)$$

**DTR-dependent estimates.** To account for diurnal temperature variation, we applied model 4 (Table 1) with the Tiefora temperature data to predict the $EIP_{50}$ of mosquitoes that fully fed at the beginning of each hour, $h$, for each date, $d$, (denoted $EIP_{DTR-dependent,h,d}$) and the HMTP of mosquitoes that fed at 0.1 hour increments. These HMTP values were used to calculate the mean hourly HMTP value for each date, which we denote $c_{DTR-dependent,h,d}$.

To account for differences in the times that mosquitoes bite throughout the night, we used data on the hourly indoor and outdoor biting times of *A. gambiae* mosquitoes in two villages in the Cascades region of Burkina Faso between 19:00 and 05:00 h[43]. We aggregated these data by village and date, and calculated the proportion mosquitoes blood-feeding during each hour (Fig. S7). We denote the empirical proportion of mosquitoes blod-feeding between hour $h$ and $h + 1$ as $B_h$; for times without mosquito collections we assumed $B_h = 0$. We assume (1) that all bites are taken in equally infectious individuals and (2) each mosquito then remains in the same location throughout the duration of infection. For each time, $t$, we specify the date at time $t$ as $d_t$. We calculate the daily values for each date at time $t$ as the weighted average given the differences in feeding times,

$$EIP_{DTR-dependent}(t) = \sum_{h=0}^{24} EIP_{DTR-dependent,h,d_t} B_h,$$

$$c_{DTR-dependent}(t) = \sum_{h=0}^{24} c_{DTR-dependent,h,d_t} B_h. \quad (4)$$

## Malaria transmission model

To estimate the effects of the temperature-dependent seasonality in the EIP, HMTP and the per-capita mosquito mortality rate, on the entomological malaria risk, we now embed our model of of the sporogonic cycle into a widely adapted model of malaria transmission dynamics: the Ross-Macdonald model[39,46] thus,

$$x = \frac{X}{N},$$
$$M = S + \sum_{i=1}^{\alpha} Y_i + Z,$$
$$z = Z/M,$$
$$\beta_j(t) = \frac{\alpha}{EIP_j(t)},$$
$$\frac{dX}{dt} = abMz(N - X) - rX,$$
$$\frac{dS}{dt} = o(t)M - ac_j(t)xS - g(t)S, \quad (5)$$
$$\frac{dY_1}{dt} = ac_j(t)xS - \beta_j(t)Y_1 - g(t)Y_1,$$
$$\dots,$$
$$\frac{dY_\alpha}{dt} = \beta_j(t)Y_{(\alpha-1)} - \beta_j(t)Y_\alpha - g(t)Y_\alpha,$$
$$\frac{dZ}{dt} = \beta_j(t)Y_\alpha - g(t)Z,$$
$$j \in [DTR - independent, DTR - dependent].$$

Model parameters are outlined in Table 2. $X$ is the number of infected people and $M$ is the number of adult female mosquitoes per person. In this model, it is assumed that the human population is constant in size. The biting rate, $a$, is assumed constant, meaning the gonotrophic cycle length is assumed to be exponentially distributed with a mean of $\frac{1}{a}$[84]. 

We allowed time-dependent changes in the parasite development rate, $\beta$, the HMTP, $c$, according to the previously described (1) DTR-dependent methods, (2) DTR-independent methods or (3) the mean values of DTR-dependent estimates (resulting in time-independent, constant parameter estimates). The per-mosquito mortality rate, $g$, was also modelled as a

**Table 2 | Malaria transmission model parameters**

| Parameter | Description |
|---|---|
| $a$ | Erlang distributed EIP shape parameter |
| $\beta$ | Erlang distributed EIP rate parameter; per day. Calculated as $\frac{\alpha}{EIP}$ |
| $g$ | Per-mosquito mortality rate; per day |
| $a$ | Per-mosquito biting rate; per day |
| $c$ | Human-to-mosquito transmission probability |
| $b$ | Mosquito-to-human transmission probability |
| $r$ | Per-human recovery rate; per day |
| $M$ | Vector-host ratio; per person |
| $o$ | Per-mosquito adult emergence rate; per day |

function of the temperature according to the Martens 2 model[6],

$$\mu(C) = e^{-\frac{1}{-4.4+1.31C-0.03C^2}},$$
$$g(t) = -\ln(\mu[\bar{C}_i(t)]), \quad (6)$$
$$i \in \{d, m\},$$

where $\mu(C)$ is daily survival probability at temperature $C$, and $\bar{C}_d(t)$ is the mean daily ($d$) or monthly ($m$) temperature at time $t$. Diurnal temperature variation can affect the mosquito survival in the laboratory[31] and mark-release-recapture studies suggest the DTR may be important[85], but we do not have a validated model of the effects in the field. For the DTR-dependent estimates we therefore calculated the values for each time, $t$, as $g(t) = -\ln(p(\bar{C}_d[t]))$, and for the DTR-independent estimates, $g(t) = -\ln(p(\bar{C}_m[t]))$.

To simulate seasonal changes in the mosquito abundance per person, $M$, we assumed $M$ is directly proportional to the sampled human biting rate, *HBR*, from nightly human landing catches in Tiefora between 2016 and 2018[69]. We fit a generalised additive models (GAM) to these data,

$$\ln(\mathbb{E}(HBR|\omega)) = \tau_0 + f(\omega) + L, \quad (7)$$

where $f$ is a univariate cyclic cubic spline with 10 knots for the day of the year, $\omega \in \{1, \dots, 365\}$ and $L$ is the location (indoors or outdoors) where mosquitoes were sampled, using a negative binomial likelihood for the total count of mosquitoes that attempted to feed on a single person;

$$HBR \sim NB(mean = \mathbb{E}(HBR|\omega), \sigma_{HBR}). \quad (8)$$

Models were fit in Stan[86], with ~ *normal*(0, 5) priors on the intercept, ~ *exp*(1) priors on the spline parameters and negative binomial over-dispersion parameter. The model was run with four chains with 4000 iterations each (inclusive of 2000 iterations warm up). Convergence was determined by $\hat{R} < 1.01$ for all parameters and visual checking of trace plots. To account for mosquito feeding, we the specified that,

$$M(t) = \frac{\mathbb{E}(HBR|\omega_t)}{a} \quad (9)$$

where $\omega_t$ is the day of the year at time $t$. To simulate changes in $M$ in model (5) according to the smoothed seasonal trend in the indoor HBRs, we estimated the the adult mosquito emergence rate required for each day, $o(t)$, given a constant per-capita daily emergence and mortality rate,[87]

$$\frac{dM}{dt} = R(t)M,$$
$$R(t) = o(t) - g(t) = \ln(M_{t+1}) - \ln(M_t), \quad (10)$$
$$o(t) = R(t) + g(t),$$

and fed $o(t)$ into the model. Note that for times within the same day $o$ is constant. We fitted the model with the DTR-dependent, DTR-independent and constant parameters (equation (5)) independently to the sampled sporozoite prevalence values ($\widehat{z}$) at each time point, by scaling the $c$ parameter in equation (5): $c_j(t)\psi$. There are numerous differences between human-to-mosquito infection in SMFAs and the field; $\psi$, therefore, represents the quantity by which the HMTP must be multiplied in the transmission model to scale the predicted sporozoite prevalence accordingly. $\psi$ was fitted using maximum likelihood with a binomial likelihood

$$\widehat{z}_t \sim B(HBR_t, z(t)), \tag{11}$$

with upper and lower bounds such that $0 \leq c(t)\psi \leq 1$. $a$ and $r$ were both assumed to be constant, and the model with each parameter set (DTR-dependent, DTR-independent or constant) was fitted independently, with a range of values for the $a$ and $r$ parameters.

To calculate the predictive accuracy of the model we separated the data into a training (years 2016 and 2017) and testing dataset (year 2018) and using the testing data calculated the AUC (calculated using the pROC R package[79]) to assess the ability of the model to predict whether individual mosquitoes are infectious and the MAE to assess the accuracy of the population level sporozoite prevalence.

### Reporting summary
Further information on research design is available in the Nature Portfolio Reporting Summary linked to this article.

### Data availability
This paper primarily uses previously published data from[30,35,40,69]. Novel data for the effects of DTR on sporogony and temperature data are available on the GitHub[88].

### Code availability
All code is available from https://github.com/IsaacStopard/EIP_HMTP_DTR[88].

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

## Acknowledgements

We thank Nakul Chitnis and Maria-Gloria Basáñez for helpful feedback. This work was supported by Wellcome Trust (226072/Z/22/Z) with novel laboratory data funded by NIH NIAID grant #R01AI110793 and National Science Foundation Ecology and Evolution of Infectious Diseases grant (DEB-1518681). I.J.S. was partly funded by the Natural Environment Research Council (NE/P012345/1) administered through the Quantitative Methods in Ecology and Evolution Centre for Doctoral Training. I.J.S. & T.S.C. acknowledge funding from the MRC Centre for Global Infectious Disease Analysis (reference MR/X020258/1), jointly funded by the UK Medical Research Council (MRC) and the UK Foreign, Commonwealth & Development Office (FCDO), under the MRC/FCDO Concordat agreement and is also part of the EDCTP2 programme supported by the European Union. AS was supported by Wellcome Trust [200222/Z/15/Z] for field mosquito collection and laboratory works in Burkina Faso.

## Author contributions

I.J.S. contributed to the study conceptualisation, funding acquisition, methodology, formal analysis and writing (original draft and editing). A.S., W.M.G., N.S., E.S. and M.B.T. contributed to the methodology, investigation and writing (review and editing). L.J.C. contributed to the supervision and writing (original draft and editing). T.S.C. and B.L. contributed to the study conceptualisation, methodology, supervision and writing (original draft and editing).

## Competing interests

The authors declare no competing interests.
