## [Transparent Peer Review file · Communications Biology]

Modelling the effects of diurnal temperature variation on malaria infection dynamics in mosquitoes

Corresponding Author: Dr Isaac Stopard

Version 0:

Reviewer comments:

Reviewer #1

(Remarks to the Author)
Paper Review Comment

Titled: Modelling the effects of diurnal temperature variation on malaria infection dynamics in mosquitoes

The paper addresses a critical aspect of malaria transmission modelling, particularly the role of temperature variation in the dynamics of Plasmodium parasite development within mosquitoes. The study's attempt to bridge the gap between constant temperature experiments and real-world temperature fluctuations offers valuable insights for enhancing the accuracy of climate-sensitive malaria transmission models.

Overall, the paper makes an important contribution to the field of malaria transmission modelling by addressing the effects of diurnal temperature variation. While the study offers valuable insights, further exploration of the model's limitations and potential applications could enhance its relevance and impact. Addressing the comments below could lead to a more comprehensive understanding of the role temperature plays in malaria transmission dynamics and inform more effective public health strategies.

1. Clarification of Model Limitations: The abstract notes that incorporating fine-scale temperature effects did not dramatically improve the model's ability to predict sporozoite prevalence in Burkina Faso. It would be beneficial to discuss why this might be the case and explore the limitations of the model in more detail. Are there other factors at play that could explain the observed discrepancies between the model predictions and empirical data?
2. Exploration of Simplified Approaches: Given the finding that increased complexity in temperature modelling did not substantially improve predictive accuracy, it would be useful to discuss whether simpler approaches could yield comparable results. This could include a comparison of the model's performance with and without the inclusion of fine-scale temperature variations.
3. Potential for Broader Application: While the study focuses on a specific site in Burkina Faso, the findings could have broader implications for other regions with different climatic conditions. Discussing how the model could be adapted or applied to other endemic areas would enhance the paper's impact and relevance.
4. Implications for Public Health Policy: The study's findings could have implications for public health interventions aimed at controlling malaria. A brief discussion on how these insights might influence vector control strategies or the timing of interventions in areas with fluctuating temperatures would be valuable.

Reviewer #3

(Remarks to the Author)

The paper discusses a study on how the varying daily temperatures affect malaria transmission by mosquitoes. It evaluates whether mathematical models, which traditionally use constant temperature data, can accurately predict malaria dynamics under fluctuating temperatures. Findings from laboratory experiments with mosquitoes exposed to varying temperatures indicate that incorporating these temperature variations can change key parameters like the time it takes for mosquitoes to become infectious and the likelihood of them transmitting malaria to humans. However, when these detailed temperature effects were included in a broader malaria transmission model, they did not significantly improve the prediction of actual infection rates in a high-transmission area in Burkina Faso. The research suggests that while temperature fluctuations do

affect laboratory measurements, they may not be crucial for predicting malaria transmission in real-world settings where other factors like human immunity and intervention strategies play significant roles.

A mathematical model is developed to assess the impacts of diurnal temperature variations on malaria infection dynamics within mosquitoes. This model is designed to predict two critical factors: the human-to-mosquito transmission probability (HMTP) and the extrinsic incubation period (EIP), under conditions of fluctuating temperatures.

The model incorporates temperature-dependent parameters derived from empirical data on thermal performance curves (TPCs) from previous constant temperature studies. These TPCs describe how key traits of malaria vectors and parasites change with temperature. The model extends these curves to account for realistic diurnal temperature variations, aiming to predict how these fluctuations influence the sporogonic cycle of the Plasmodium parasites within mosquitoes.

Once the model estimates the HMTP and EIP under varied temperature profiles, it embeds these parameters into a broader malaria transmission dynamics model. This overarching model simulates the prevalence of sporozoites (the infectious form of the parasite) in mosquito populations and examines how well these predictions align with observed data from field studies, particularly focusing on a site in Burkina Faso.

In summary, the model is a sophisticated tool that bridges laboratory-derived thermal performance data with dynamic environmental temperature conditions to better understand and predict malaria transmission patterns under more realistic, variable climatic scenarios.

The model relies heavily on laboratory data, which might not capture the full complexity of natural environments where multiple factors affect malaria transmission, not just temperature variations. Despite innovative attempts to include these temperature fluctuations, the model didn't significantly improve predictions in real-world settings, suggesting that other ecological and human factors are crucial and need to be integrated into the model for more accurate predictions. However, the model integrates laboratory data with real-world environmental conditions, specifically by factoring in diurnal temperature variations. This approach enhances understanding of how temperature affects malaria transmission in mosquitoes. It also improves predictions related to key transmission parameters like the human-to-mosquito transmission probability and the extrinsic incubation period. As the authors say 'Irrespective of whether these behaviours are caused by temperature, the framework we have developed could be used to predict mosquito infection in different scenarios'. I see this work as a step in the correct direction.

Version 1:

Reviewer comments:

Reviewer #4

(Remarks to the Author)

The authors have thoroughly addressed all the questions raised by the two reviewers, ensuring that each point was carefully considered and incorporated into the revised manuscript where appropriate.

Reviewers' comments:

Reviewer #1 (Remarks to the Author):

Paper Review Comment

Titled: Modelling the effects of diurnal temperature variation on malaria infection dynamics in mosquitoes

The paper addresses a critical aspect of malaria transmission modelling, particularly the role of temperature variation in the dynamics of Plasmodium parasite development within mosquitoes. The study's attempt to bridge the gap between constant temperature experiments and real-world temperature fluctuations offers valuable insights for enhancing the accuracy of climate-sensitive malaria transmission models. Overall, the paper makes an important contribution to the field of malaria transmission modelling by addressing the effects of diurnal temperature variation. While the study offers valuable insights, further exploration of the model's limitations and potential applications could enhance its relevance and impact. Addressing the comments below could lead to a more comprehensive understanding of the role temperature plays in malaria transmission dynamics and inform more effective public health strategies.

Thank you very much for this summary, positive comments and constructive feedback.

1. Clarification of Model Limitations: The abstract notes that incorporating fine-scale temperature effects did not dramatically improve the model's ability to predict sporozoite prevalence in Burkina Faso. It would be beneficial to discuss why this might be the case and explore the limitations of the model in more detail. Are there other factors at play that could explain the observed discrepancies between the model predictions and empirical data?

This is a good point. Although the model of mosquito infection in standard membrane feedings assays better predicted the differences in the controlled laboratory conditions (Figure 1), the predicted parameter differences for the DTR-dependent and DTR-independent models using the ERA5 temperature data (Figure 3) were sometimes small, meaning it was difficult to distinguish between the models against field data (Figure 5). We currently have a long list of possible limitations in the discussion (i.e. need for more settings, transmission dynamics model, circadian rhythms in the human, vector control practices, differences in mosquito mortality with temperature, mosquito resting behaviour) but we agree there are others not mentioned. We have therefore modified the discussion as follows,

“For temperatures lower or greater than the experimental constant temperatures, we assumed the PDR was the lowest or highest values respectively. Experimental data for the impact of DTR at higher temperatures are lacking but would be required to further validate this model (the maximum experimental temperature applied was 32°C within the 27°C DTR experiment). Nonetheless, this method improves the model fits of laboratory data and replicates the observation that the EIP of mosquitoes maintained at 19°C with a DTR of 10°C is shorter than those with a DTR of 0°C, whilst there is little difference at 27°C. The magnitudes of difference are also closely mirrored. Existing EIP estimates suggest that the PDR does not decline to zero at temperatures below 35°C.”

Because experimental data exists where the EIP did not decline up to temperatures of at least 35°C we do not believe this assumption and the ERA5 temperatures were rarely greater than 35°C (Figure 3) we do not believe this limitation substantially affected our results. We have clarified this by adding,

“The ERA5 temperatures were rarely greater than 35°C, so we do not believe our assumptions regarding the limits of PDR and HMPD substantially affected our parameter estimates for *Tiefora*. Whether the ERA5 data is a good predictor of temperatures experienced by the mosquito is, however, unclear, as our empirical temperature estimates showed considerably more variability than ERA5”.

Finally, the reliability of the ERA5 data could be questioned together with the robustness of the sporozoite prevalence data. So we included the following sentences,

" The ERA5 temperatures were rarely greater than 35°C, so we do not believe our assumptions regarding the limits of PDR and HMPD substantially affected our parameter estimates for *Tiefora*. Whether the ERA5 data is a good predictor of temperatures experienced by the mosquito is, however, unclear, as our empirical temperature estimates showed considerably more variability than ERA5. Sporozoite prevalence data is highly variable and large numbers of mosquitoes are required to differentiate between models unless predictions vary substantially. As previously outlined, the predictions of the DTR-dependent and DTR-independent model were relatively similar in this setting, so the power of the study to differentiate between models is likely to be relatively low. Given the substantial limitations outlined above, we would argue that further research is needed to verify whether daily fluctuations in temperature influence malaria transmission in wild mosquito populations. Understanding discrepancies between the laboratory and different field settings therefore remains critical. Once validated, the framework outlined here could be used to predict mosquito infection in different scenarios and explore how temperature may influence the effectiveness of malaria control interventions in changing environments"

2. Exploration of Simplified Approaches: Given the finding that increased complexity in temperature modelling did not substantially improve predictive accuracy, it would be useful to discuss whether simpler approaches could yield comparable results. This could

include a comparison of the model's performance with and without the inclusion of fine-scale temperature variations.

Many thanks – this is a great point and an important question. We feel, however, we have already addressed this by including models without the effects of temperature and models without the effects of short-term temperature fluctuations (DTR-independent). For example, we include a comparison of the simpler DTR-independent model and more complex DTR-dependent model performance in Figure 5. In this setting there was not much difference between the models, making it hard to differentiate between the two hypotheses. We have tried to reinforce this in the point above and by rewording the abstract to make this point more explicit.

“Including the effects of diurnal temperature variation did not improve the ability of the transmission model to predict changes in sporozoite prevalence in wild mosquitoes, and further work is needed in a wider range of settings to understand how daily fluctuating temperatures might impact malaria transmission.”

3. Potential for Broader Application: While the study focuses on a specific site in Burkina Faso, the findings could have broader implications for other regions with different climatic conditions. Discussing how the model could be adapted or applied to other endemic areas would enhance the paper's impact and relevance.

Thank you, we agree this is an important point. We have edited the discussion to include the following details,

“We attempted to validate the laboratory data and our framework for estimating the EIP and HMTP in a fluctuating environment by including the parameter estimates in a simple transmission dynamics model, and predicting sporozoite prevalence for a single site in Burkina Faso with a specific set of temperatures. The temperature-dependent malaria transmission models were broadly able to capture seasonality.”

“Further data from a range of sites with different temperature profiles is needed to validate the system and determine whether malaria models should include diurnal temperature range dependencies in the EIP and HMTP.”

4. Implications for Public Health Policy: The study's findings could have implications for public health interventions aimed at controlling malaria. A brief discussion on how these insights might influence vector control strategies or the timing of interventions in areas with fluctuating temperatures would be valuable.

This is an interesting point and one we have not considered. Our findings could help improve mechanistic model predictions of how malaria transmission will change with temperature, which in turn could assist in the allocation of malaria control. Currently the

major vector control intervention is the use of insecticide treated nets which last for multiple years, so it is unclear how changes in the timing of their distribution would impact control. As more methods of control are developed (for example, spatial repellents) this might become more important. However, we feel that further validation in field settings is needed prior to the voicing of policy recommendations so though we will certainly consider it in future. To emphasise this we have added the following section to the discussion.

“Once validated, the framework outlined here could be used to predict mosquito infection in different scenarios, explore how temperature may influence the effectiveness of malaria control interventions in changing environments and incorporated into decisions tools for malaria intervention deployment.”

Reviewer #3 (Remarks to the Author):

The paper discusses a study on how the varying daily temperatures affect malaria transmission by mosquitoes. It evaluates whether mathematical models, which traditionally use constant temperature data, can accurately predict malaria dynamics under fluctuating temperatures. Findings from laboratory experiments with mosquitoes exposed to varying temperatures indicate that incorporating these temperature variations can change key parameters like the time it takes for mosquitoes to become infectious and the likelihood of them transmitting malaria to humans. However, when these detailed temperature effects were included in a broader malaria transmission model, they did not significantly improve the prediction of actual infection rates in a high-transmission area in Burkina Faso. The research suggests that while temperature fluctuations do affect laboratory measurements, they may not be crucial for predicting malaria transmission in real-world settings where other factors like human immunity and intervention strategies play significant roles.

A mathematical model is developed to assess the impacts of diurnal temperature variations on malaria infection dynamics within mosquitoes. This model is designed to predict two critical factors: the human-to-mosquito transmission probability (HMTP) and the extrinsic incubation period (EIP), under conditions of fluctuating temperatures.

The model incorporates temperature-dependent parameters derived from empirical data on thermal performance curves (TPCs) from previous constant temperature studies. These TPCs describe how key traits of malaria vectors and parasites change with temperature. The model extends these curves to account for realistic diurnal temperature variations, aiming to predict how these fluctuations influence the sporogonic cycle of the Plasmodium parasites within mosquitoes.

Once the model estimates the HMTP and EIP under varied temperature profiles, it embeds these parameters into a broader malaria transmission dynamics model. This overarching model simulates the prevalence of sporozoites (the infectious form of the parasite) in mosquito populations and examines how well these predictions align with observed data

from field studies, particularly focusing on a site in Burkina Faso.

In summary, the model is a sophisticated tool that bridges laboratory-derived thermal performance data with dynamic environmental temperature conditions to better understand and predict malaria transmission patterns under more realistic, variable climatic scenarios.

The model relies heavily on laboratory data, which might not capture the full complexity of natural environments where multiple factors affect malaria transmission, not just temperature variations. Despite innovative attempts to include these temperature fluctuations, the model didn't significantly improve predictions in real-world settings, suggesting that other ecological and human factors are crucial and need to be integrated into the model for more accurate predictions.

However, the model integrates laboratory data with real-world environmental conditions, specifically by factoring in diurnal temperature variations. This approach enhances understanding of how temperature affects malaria transmission in mosquitoes. It also improves predictions related to key transmission parameters like the human-to-mosquito transmission probability and the extrinsic incubation period. As the authors say 'Irrespective of whether these behaviours are caused by temperature, the framework we have developed could be used to predict mosquito infection in different scenarios'. I see this work as a step in the correct direction.

Thank you very much for this accurate description. We have reduced our emphasis on the difficulties in predicting the field data in the abstract, as we believe that diurnal temperature fluctuations are important in the laboratory, but we might not be able to accurately predict them in the field.

We have also added further information into the introduction,

“Evidence from other species indicates performance in fluctuating temperatures might not correspond directly to the corresponding instantaneous temperatures due to differences in phenotypic plasticity and acclimation, and hysteresis effects due to damage and repair from exposure to thermal extremes.”